# Untangling the effects of multiple exposures with a common reference group in an epidemiologic study: A practical revisit

**Robert E. Fontaine[1], Yulei He[2], Bao-Ping Zhu[3]***

1 Division of Global Health Protection, Global Health Center, Centers for Disease Control and Prevention, Atlanta, Georgia, United States of America, 2 Division of Research Methodology, National Center for Health Statistics, Centers for Disease Control and Prevention, Hyattsville, Maryland, United States of America, 3 Office of Science Quality and Library Services, Office of Science, Centers for Disease Control and Prevention, Atlanta, Georgia, United States of America

* bzhu@cdc.gov

**Data Availability Statement:** All relevant data are within the paper and its Supporting Information files.

## Abstract

When assessing multiple exposures in epidemiologic studies, epidemiologists often use multivariable regression models with main effects only to control for confounding. This method can mask the true effects of individual exposures, potentially leading to wrong conclusions. We revisited a simple, practical, and often overlooked approach to untangle effects of the exposures of interest, in which the combinations of all levels of the exposures of interest are recoded into a single, multicategory variable. One category, usually the absence of all exposures of interest, is selected as the common reference group (CRG). All other categories representing individual and joint exposures are then compared to the CRG using indicator variables in a regression model or in a 2×2 contingency table analysis. Using real data examples, we showed that using the CRG analysis results in estimates of individual and joint effects that are mutually comparable and free of each other's confounding effects, yielding a clear, accurate, intuitive, and simple summarization of epidemiologic study findings involving multiple exposures of interest.

## Introduction

Epidemiologists frequently use case-control, retrospective cohort, and cross-sectional studies to assess the effects of multiple exposures on a disease in a single study. When analyzing data from such studies, researchers frequently start by comparing each group exposed to an exposure of interest with the remainder without that same exposure. However, such analyses lead to the "shifting reference group" problem [1], where the effect measures (e.g., risk or odds ratios) for the exposures are not mutually comparable because the denominators differ. The effect estimates also trend toward the null value because each exposure is compared to a mixture of other exposures. To address problems with multiple exposures, researchers often use a multivariable regression model to control for confounding. However, this approach does not resolve the problem with shifting reference group, contrary to common belief.

**Funding:** The author(s) received no specific funding for this work.

**Competing interests:** The authors have declared that no competing interests exist.

Conversely, using a common reference group (CRG) for each individual and joint exposure cleanses the problems of shifting reference groups. The CRG may be the group lacking all exposures of interest, the one with the lowest risk, or one selected by the analyst to be the appropriate reference standard [2]. Once the CRG has been chosen, all joint and individual effects are separated, and each is individually compared to the CRG.

Use of a CRG when comparing the individual effects of two or more exposures has been mentioned in several textbooks, without further elaboration [1–5]. Some texts limit the discussion to assessing interaction [6–8]. Similarly, while several journal articles since 2000 have provided coverage on the use of a CRG in the context of determining additive interaction [8–14], we could find only one that discussed estimating, comparing, or disentangling the individual effects in the context of gene-environment interaction [14]. A review of 138 studies published from 2001 to 2007 showed that 89% did not apply a CRG to estimate the individual and joint effects when multiple exposures were involved [13]. Continued underuse of the CRG may lead to missed opportunities to identify important exposures as well as misleading quantification and comparison of effect estimates [10, 13, 14].

In our practice, we frequently see epidemiologic analyses with the problem of the shifting reference group, potentially leading to misunderstanding and misinterpretation of results. Accordingly, in this paper we aim to provide a practical, straightforward illustration of the CRG approach. We provide real data examples, lay out practical approaches to data analysis, provide an intuitive illustration of the underlying methodological issues, and discuss advantages and caveats in using the CRG approach. We advocate for the use of the CRG as a standard approach in analyzing epidemiologic studies involving multiple exposures.

## Materials and methods

### Typical approach in epidemiologic data analysis

Suppose in a study there is a binary outcome Y, with values of 1 or 0 (e.g., Y = 1 for being ill and Y = 0 for being well). There are also three binary exposures, $X_1$, $X_2$, and $X_3$, each taking on values 1 or 0 (exposed vs. nonexposed). Typically, the epidemiologic analysis is conducted as follows.

Step 1. Univariate analysis: The relationships, Y vs. $X_1$, Y vs. $X_2$, and Y vs. $X_3$, are evaluated separately, using either three 2×2 tables or univariate logistic regression models.

$$Logit(P(Y = 1)) = \beta_{o1} + \beta_1 X_1,$$

$$Logit(P(Y = 1)) = \beta_{o2} + \beta_2 X_2,$$

$$Logit(P(Y = 1)) = \beta_{o3} + \beta_3 X_3,$$

where $Logit(P(Y = 1))$ is the logit link function for modeling the probability of Y = 1 (e.g., being ill); $\beta_o$ is the intercept of the regression line; $\beta_i$ (for i = 1,2,3) is the regression coefficient corresponding to the $i^{th}$ exposure factor, $X_i$.

In a 2×2 table or univariate regression model, the group with exposure to $X_1$, ($X_1 = 1$), is compared to a reference group consisting of those with $X_1 = 0$ and a mixture of those with $X_2 = 0$, $X_2 = 1$, $X_3 = 0$, and $X_3 = 1$. Similarly, the group with exposure to $X_2$ ($X_2 = 1$), is compared to a reference group consisting of those with $X_2 = 0$ and a mixture of those with $X_1 = 0$, $X_1 = 1$, $X_3 = 0$, and $X_3 = 1$; and so on. Hence each effect estimate has a different ("shifting") reference group. Since each reference group contains the other exposures, the effect measures will be biased toward the null. This is akin to an experiment with multiple experimental arms, in which the researcher compares an experimental arm to the combination of other experimental

arms plus the control arm, instead of comparing each individual experimental arm to the control arm.

Step 2. Multivariate analysis with main effects: Suppose $X_1$, $X_2$ and $X_3$ from step 1 are all associated with outcome, $Y$. The main-effect multivariable regression model is often used to control for confounding of those risk factors simultaneously:

$$Logit(P(Y = 1)) = \beta_o + \beta_1 X_1 + \beta_2 X_2 + \beta_3 X_3$$

Epidemiologists commonly assume that the multivariable regression model takes care of the shifting reference group problem, when in fact, it does not. For example, from the main-effect logistic regression model, the adjusted odds ratio ($OR_{adj}$) associated with $X_1 = 1$ $vs.$ $X_1 = 0$, $OR_{adj}$ ($X_1 = 1$ $vs.$ $X_1 = 0$), controlling for $X_2$ and $X_3$, is calculated as follows:

$$OR_{adj}(X_1 = 1 \ vs. X_1 = 0) = \frac{\exp(\beta_o + \beta_1 \cdot 1 + \beta_2 X_2 + \beta_3 X_3)}{\exp(\beta_o + \beta_1 \cdot 0 + \beta_2 X_2 + \beta_3 X_3)} = \exp(\beta_1).$$

Similarly, $OR_{adj}$ ($X_2 = 1$ $vs.$ $X_2 = 0$) and $OR_{adj}$ ($X_3 = 1$ $vs.$ $X_3 = 0$), controlling for other exposures, are calculated as follows:

$$OR_{adj}(X_2 = 1 \ vs. X_2 = 0) = \frac{\exp(\beta_o + \beta_2 \cdot 1 + \beta_1 X_1 + \beta_3 X_3)}{\exp(\beta_o + \beta_2 \cdot 0 + \beta_1 X_1 + \beta_3 X_3)} = \exp(\beta_2),$$

$$OR_{adj}(X_3 = 1 \ vs. X_3 = 0) = \frac{\exp(\beta_o + \beta_3 \cdot 1 + \beta_1 X_1 + \beta_2 X_2)}{\exp(\beta_o + \beta_3 \cdot 0 + \beta_1 X_1 + \beta_2 X_2)} = \exp(\beta_3).$$

Notice the denominators, i.e., $\exp(\beta_o + \beta_1 \cdot 0 + \beta_2 X_2 + \beta_3 X_3)$, $\exp(\beta_o + \beta_2 \cdot 0 + \beta_1 X_1 + \beta_3 X_3)$ and $\exp(\beta_o + \beta_3 \cdot 0 + \beta_1 X_1 + \beta_2 X_2)$, differ from each other in terms of exposures; hence the problem of shifting reference group persists. The same can be extended to other generalized linear models (GLMs) for different types of outcome variables where there are at least two exposures. This point will be further illustrated in the numeric examples section.

Several consequences result from the shifting reference group problem. First, the effect measures are no longer comparable to one another because each has a different reference level. Second, the shifting reference group frequently biases association measures toward the null [1], hence reducing the probability of identifying a statistically significant exposure and lessening the magnitude of the effect estimate. Third, patterns of interest in the joint effect of two or more variables may be missed.

## The common reference group (CRG) approach

The CRG approach should be considered when two or more exposures, $X_1$, $X_2$, $X_3$... may be relevant in an investigation. This may arise from the hypothesis, the study question, or *a priori* knowledge of the exposure-disease relationship. During an epidemiologic study, the interplay of multiple exposures may first be noticed during the initial descriptive analysis. For example, foodborne disease outbreak investigations often begin with the observations that only people eating meals at the same time and place became ill, and that several foods have relatively weak associations with illness that have or approach statistical significance. A CRG approach would be needed to untangle the effects of these foods. Even when the effect measures are high, a CRG approach will still help sort them out by degree of importance.

The next step involves recoding the exposure variables, $X_1$, $X_2$, $X_3$, into a single, new exposure variable, $Z$, the values of which are a combination of values of exposure variables, $X_1$, $X_2$, and $X_3$ (Table 1). The recoding can be done using a spreadsheet, database, or statistical software.

**Table 1. Illustrative example of creating a common reference group (CRG) by recoding exposures, $X_1$, $X_2$, and $X_3$.**

| Group | $X_1$ | $X_2$ | $X_3$ | Factors exposed to | Z |
|---|---|---|---|---|---|
| A | 1 | 1 | 1 | $X_1$, $X_2$, and $X_3$ | 8 |
| B | 1 | 1 | 0 | $X_1$, $X_2$ but not $X_3$ | 7 |
| C | 0 | 1 | 1 | $X_2$, $X_3$ but not $X_1$ | 6 |
| D | 1 | 0 | 1 | $X_1$, $X_3$ but not $X_2$ | 5 |
| E | 1 | 0 | 0 | $X_1$, but not $X_2$, $X_3$ | 4 |
| F | 0 | 1 | 0 | $X_2$, but not $X_1$, $X_3$ | 3 |
| G | 0 | 0 | 1 | $X_3$, but not $X_1$, $X_2$ | 2 |
| H (CRG) | 0 | 0 | 0 | None of the three | 1 |

In the analysis, variable Z is used instead of variables $X_1$, $X_2$, and $X_3$. Using the standard 2×2 table analysis, each group, A–G, is individually compared with the CRG, Group H (i.e., A vs. H, B vs. H, etc.). One may also use a regression analysis such as logistic regression, in which variable Z is recoded into indicator variables, and Group H serves as the CRG. For large samples, both methods yield mathematically identical results; for small samples, however, regression analyses may result in matrix tolerance problems due to zero or sparse cells.

Once recoded, the effect measure can be computed by comparing each of the exposure groups, A–G, to the CRG, H (i.e., A vs. H, B vs. H, etc.), using a 2×2 table analyses. Alternatively, one may use logistic regression (or other GLMs) with variable Z being recoded to indicator variables.

The CRG may be any logically appropriate group chosen by the epidemiologist. Usually, it is the group with absence of the component exposures [4], i.e., group H in Table 1. However, not all situations have an absence-of-exposure group. For example, when the suspected exposure is in the drinking water during an outbreak, the CRG cannot be a group that did not drink any water. Instead, the CRG can be the exposure group with the lowest risk in a univariate analysis [15, 16]. When assessing the effectiveness of multiple protective measures (e.g., facemask use, handwashing, air filtration, physical distancing) for preventing respiratory infection, the group with exposure to the infection and without any of these protective measures may serve as the CRG. Alternatively, one may compute risk scores using the number of protective measures and create a group with the lowest risk score as the CRG. Groups with higher risk scores can be compared to the CRG to assess the dose-response relationship. Examples of these uses of CRG approach include an investigation of risk factors for influenza [17], and a study of the preventive effect of handwashing on hand, foot, and mouth disease and herpangina [18].

A mathematically equivalent approach to the CRG is to force all possible interaction terms into a saturated multiple regression model [19]. This approach has practical difficulties and drawbacks, which are discussed later in this paper. Finally, adjusting for potential confounding effects of other variables, such as age or socioeconomic status, may still be needed in the data analysis.

## Results

Here we use two data examples to show how using a CRG can lead to richer and more illuminating study findings.

### Ebola virus disease outbreak

Investigation of a large Ebola virus disease outbreak in Zaire in 1976 provides a simple example of two exposures of interest [20, 21]. From the initial descriptive epidemiologic analysis, the

**Table 2. Case-control study of risk factors for Ebola virus disease, Zaire, 1976: Crude analysis, main-effect logistic regression, and common reference group (CRG) analysis.**

| | Univariate and Main-Effect Logistic Regression Analysis | | | | CRG Analysis | | | | |
|---|---|---|---|---|---|---|---|---|---|
| Exposure* | Cases (N = 318) | Controls (N = 318) | $OR_{crude}$ (95% CI)† | $OR_{adj}$ (95% CI)‡ | Hosp‖ | Comm¶ | Cases (N = 318) | Controls (N = 318) | $OR_{CRG}$ (95% CI)§ |
| Hosp‖ (+) | 128 | 26 | 7.5 (4.8–12) | 29 (18–48) | (+) | (+) | 43 | 4 | 70 (24–205) |
| Hosp‖ (-) | 190 | 292 | Ref | Ref | (+) | (-) | 85 | 22 | 25 (14–44) |
| Comm¶ (+) | 192 | 30 | 15 (9.5–23) | 19 (11–32) | (-) | (+) | 149 | 26 | 37 (22–63) |
| Comm¶ (-) | 126 | 288 | Ref | Ref | (-) | (-) | 41 | 266 | Ref |

OR = odds ratio, CI = confidence interval, NC = not calculated; Ref = reference level.

* Exposure (+): Exposed; (-): Not exposed.

† Crude odds ratio (OR) and 95% confidence interval (CI) from the univariate analysis.

‡ Adjusted OR and 95% CI from the main-effect logistic regression analysis.

§ OR and 95% CI from 2×2 table analysis using a common reference group.

‖ Hosp: hospital exposure, i.e., a hospital worker, visitor, or patient.

¶ Comm: community exposure, i.e., any person with face-to-face exposure to a suspected hemorrhagic fever case in the community.

investigators hypothesized that there were two distinct modes of transmission: hospital (hospital worker, visitor, or patient), and community (person-to-person contact in communities). The investigators conducted a well powered case-control study (318 cases and 318 controls) to confirm this hypothesis.

Univariate analysis of the data (**S1 File**, first tab, available in the Instructor's Guide of the TEPHINET case study [21]) showed that the disease was highly significantly associated with exposures to both the hospital (crude OR = 7.5, 95% CI: 4.8–12) and the community cases (crude OR = 15, 95% CI: 8.5–23). The adjusted ORs from a main-effects logistic regression model increased moderately for both hospital (adjusted OR = 29, 95% CI: 18–48) and community (adjusted OR = 19, 95% CI: 11–32) exposures (Table 2).

However, using those with no hospital or community exposures as the CRG, the OR for hospital-only exposure was 25, that for community-only exposure was 37, whereas that for those with both hospital and community exposures was 70 (Table 2). These ORs illustrated both the individual and joint effects of hospital and community exposures. Of note, a saturated logistic regression model would give the same ORs as the 2×4 table (not shown).

## Cholera in a refugee camp

Epidemiologists investigated an acute cholera outbreak in a refugee settlement in Uganda during 2018 to determine if drinking water was the source [15]. The refugees had three main sources of water: a stream running through the camp, a government-managed water storage tank, and a spring of groundwater. From both the univariate analysis and the main-effect logistic regression analysis of the case-control study data (**S1 File**, second tab), the ORs associated with drinking the stream water and tank water ranged from 2.2 to 2.5, and an inverse association was found with drinking the spring water (Table 3).

Using the CRG approach with eight combinations of water sources, seven were compared individually against the CRG (the eighth) of only drinking boiled, bottled, or rainwater. The analysis revealed much stronger and significant associations of cholera with drinking water from both the tank and the stream (OR = 17), the stream only (OR = 14), and from the tank only (OR = 12), compared with the univariate and main-effect logistic regression analyses (Table 3). This finding prompted further investigation, which revealed that the tank water was

**Table 3. Case-control study of risk factors for cholera, Uganda, 2018: The comparison of the crude analysis, main-effect logistic regression, and common reference group analysis.**

| | Univariate and Main-Effects Logistic Regression Analysis | | | | Common Reference Group Analysis | | | | | |
|---|---|---|---|---|---|---|---|---|---|---|
| Exp* | Cases (N = 73) | Controls (N = 107) | $OR_{crude}$ (95% CI)† | $OR_{adj}$ (95% CI)‡ | Spr* | Tank* | Str* | Cases (N = 73) | Controls (N = 107) | $OR_{CRG}$ (95% CI)§ |
| Str(+) | 48 | 46 | 2.5 (1.4–4.7) | 2.2 (1.2–4.1) | (+) | (+) | (+) | 1 | 0 | NC |
| Str(-) | 25 | 61 | Ref | Ref | (+) | (+) | (-) | 0 | 11 | 0 (0–60)‖ |
| | | | | | (-) | (+) | (+) | 39 | 36 | 17 (2.2–137) |
| Tank(+) | 64 | 80 | 2.4 (1.1–5.5) | 2.5 (1.1–5.9) | (+) | (-) | (+) | 0 | 1 | NC |
| Tank(-) | 9 | 27 | Ref | Ref | (+) | (-) | (-) | 0 | 1 | NC |
| | | | | | (-) | (-) | (+) | 8 | 9 | 14 (1.5–133) |
| Spr(+) | 1 | 13 | 0.10 (0.013–0.79) | 0.13 (0.016–1.0) | (-) | (+) | (-) | 24 | 33 | 12 (1.4–94) |
| Spr(-) | 72 | 94 | Ref | Ref | (-) | (-) | (-) | 1 | 16 | CRG¶ |

Exp = exposure, Str = stream, Spr = spring, OR = odds ratio, CI = confidence interval, CRG = common reference group, NC = not calculated; Ref = reference level.

* (+): Exposed; (-): Non-exposed.

† Crude odds ratio (OR) and 95% confidence interval (CI) from the univariate analysis.

‡ Adjusted OR and 95% CI from the main-effect logistic regression analysis.

§ OR and 95% CI from 2×2 table analysis using a common reference group.

‖ Fisher's exact confidence interval.

¶ Common reference group: Those who drank boiled, bottled, or rainwater

not drawn from the municipal drinking water system as contracted; rather, it was taken from a nearby lake. A cholera outbreak was ongoing in the lakeshore villages, where open defecation was common. Although unproven with microbiological methods, investigators suspected that the contaminated lake water, trucked to the camp storage tank introduced cholera into the camp. The unprotected stream likely was secondarily contaminated after the outbreak began. This finding more accurately informed public health interventions for controlling the outbreak.

Note that with a 2×8 table, three interaction cells were sparsely populated. The respective ORs for these cells were not calculated, but the cells were left in the table to maintain the segregation of the respective data from the other terms and assure untangling of the effect estimates for the individual sources. A saturated logistic regression model could not be fitted due to matrix tolerance being exceeded.

## Discussion

We have shown that, when multiple exposures are evaluated in an epidemiologic study, using CRG analysis leads to more accurate understanding of the relationship between the exposures and outcome. A major utility of the CRG approach is that all individual and joint effect measures are explicitly presented [10, 11, 13, 14]. They are all untangled from each other and mutually comparable. In the data examples provided above, one can observe the dramatic increases in the effect measures for competing exposures when using the CRG analysis. We have shown the theoretical argument that in regression modeling the denominators for individual exposures differ and thus, the magnitudes of their individual ORs are not comparable to each other.

The CRG analysis can be conducted in three ways that yield identical effect estimates. The first two are based on recoding the combinations of exposure levels into a 2×N table, followed by either the 2×2 table analyses in which each combination of the exposure levels is compared

with the CRG, or regression modeling of the indicator variable representing combinations of the exposure levels with the CRG serving as the reference group.

The third is to use a saturated regression model, with all possible interaction terms being forced into the model regardless of their statistical significance. However, saturated regression models are rarely used in practice for several reaons. The complexity of the saturated model increases beyond two-way interactions, requiring tedious additional effort to calculate the individual and joint effects. Increasing number of interaction terms can lead to sparsely populated cells, causing the regression model to fail due to matrix tolerance being exceeded; yet even when the interaction terms are not statistically significant or otherwise inconsequential, they need to be kept in the regression model for correct calculation of the individual and joint effects. Finally, the resultant effect measures from the saturated models are limited to assessment of multiplicative effects, yet most literature or textbooks we have surveyed that cover this topic propose the CRG (or analogous terminology) for assessing additive interaction without stressing the importance of accurately estimating the individual effects [6–14]. In contrast to the saturated regression model, the recoding provides measures of effect and impact that are intuitively interpretable as individual and joint effects. For these reasons, it is more straightfoward and convenient to present and interpret the results using the 2×N layout.

Using the CRG simulates an experiment with different intervention arms compared to a common control arm. The CRG thus provides direct estimates of the effects of individual exposures (e.g., vaccination or using personal protective equipment alone) and the effectiveness of combined measures (i.e., vaccination and using personal protective equipment).

The CRG analysis controls for confounding among the exposures of interest by restricting the analysis to one exposure at a time, thus the effects of different exposures are untangled. If one wants to consider other confounding variables, adjustment can be made by stratifying the CRG analysis by those confounders. A Mantel-Haenszel test may be applied to each comparison from the 2×N table [2]. To adjust for several confounders, the recoded single exposure variable may be used as an indicator variable in a multivariate logistic regression model and the adjustments made by including the confounders in the model.

The choice of the CRG might require additional consideration in certain situations. In the cholera outbreak example, the CRG could not be those who drank no water since everyone had to drink water. As such, not getting drinking water from any of the three main sources at the camp was chosen as the CRG [15]. Another common situation arises in food-borne disease outbreak investigations when the causative agent is distributed among several foods. This could happen from a common ingredient, cross-contamination during preparation, an infected food handler, or an infected customer at a self-serve buffet. The outbreak may be associated with several contaminated foods that may be apparently unrelated. Each will have unimpressive effect measures, some of which may not be statistically significant. The investigator could erroneously dismiss these food items as unimportant, hence missing important insights about the outbreak. In such situations, a reasonable approach might involve selecting several foods with the highest risk ratios and using the rest of the foods together as the CRG.

Recoding several exposures into a single variable can lead to sparsely populated cells. Zero disease in the CRG will lead to uninformative effective estimates (e.g., infinity risk ratios) for all individual and joint exposures. However, the disease prevalence alone for each individual and joint exposure will then equal the risk difference, a valid estimation of risk. With cross-sectional and retrospective cohort studies, the risk difference and attributable fraction will be available directly from the 2×N table. Case control or case base studies can also yield valid risk differences if the controls have a known sampling fraction from the population. Sparse, non-zero cells in the CRG can be handled by substituting a well populated group and inverting the resultant ORs accordingly. Sparse cells representing individual exposures are more

problematic since individual effects are usually more important to estimate. The workaround for sparse cells for a joint effect could be to ignore these groups altogether as was done in the cholera investigation [15] or to use statistical techniques dealing with sparse cells, such as Firth's logistic regression [22]. The sparse cell issue is not specific to the CRG approach. In anticipation of analyzing data using the CRG, epidemiologists may consider increasing the sample size and oversampling the CRG or other exposure groups of special interest based on the descriptive epidemiologic analysis.

In conclusion, a CRG approach provides effect measures that are easily interpreted. The CRG corrects the shifting reference group thus lending to comparability among effect estimates. Individual effects are also freed of confounding from other exposures of interest. The 2×N table display also provides intuitive interpretations on the effects of individual and joint exposures. The CRG approach may be combined with multivariate modelling for the stratification of confounding from extraneous variables. We, therefore, advocate the use of the CRG in retrospective epidemiologic studies whenever multiple exposures are known or suspected to be involved.

## Supporting information

**S1 File. Data sets on Ebola outbreak in Zaire (first tab) and cholera outbreak in Uganda (second tab).**
(XLSX)

## Acknowledgments

We thank Dr. Fred Monje and other colleagues at the Uganda Public Health Fellowship Program for letting us use the cholera outbreak investigation as an illustrative example and for agreeing to publish the associated data set.

**Disclaimer:** The findings and conclusions in this manuscript are those of the authors and do not necessarily represent the official position of the Centers for Disease Control and Prevention.

## Author Contributions

**Conceptualization:** Bao-Ping Zhu.

**Formal analysis:** Robert E. Fontaine, Yulei He, Bao-Ping Zhu.

**Methodology:** Robert E. Fontaine, Yulei He, Bao-Ping Zhu.

**Project administration:** Robert E. Fontaine.

**Validation:** Robert E. Fontaine, Yulei He, Bao-Ping Zhu.

**Writing – original draft:** Robert E. Fontaine, Bao-Ping Zhu.

**Writing – review & editing:** Robert E. Fontaine, Yulei He, Bao-Ping Zhu.

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
