## [Decision Letter · Decision Letter 0]

29 Aug 2023

PONE-D-23-13611Untangling the Effects of Multiple Exposures with a Common Reference Group in an Epidemiologic Study: A Practical RevisitPLOS ONE

Dear Dr. Zhu,

Thank you for submitting your manuscript to PLOS ONE. After careful consideration, we feel that it has merit but does not fully meet PLOS ONE’s publication criteria as it currently stands. Therefore, we invite you to submit a revised version of the manuscript that addresses the points raised during the review process.

Please work on your introduction, purpose of study and background by adding more precise literature. If possible use an editor to edit your work before resubmitting to avoid typo errors. Also see below comments from a reviewer and address them

We look forward to receiving your revised manuscript.

Kind regards,

Maureen Musimbi Akolo, Ph.D

Academic Editor

PLOS ONE

Journal Requirements:

Reviewers' comments:

Reviewer's Responses to Questions

**Comments to the Author**

1. Is the manuscript technically sound, and do the data support the conclusions?

Reviewer #1: Yes

2. Has the statistical analysis been performed appropriately and rigorously? 

Reviewer #1: N/A

3. Have the authors made all data underlying the findings in their manuscript fully available?

Reviewer #1: No

4. Is the manuscript presented in an intelligible fashion and written in standard English?

Reviewer #1: Yes

5. Review Comments to the Author

Reviewer #1: The authors describe method of analyzing epidemiological data using a common reference group, this was to avoid misinterpretation of data and consequential shifting reference problem that would follow.The article is well written, easy to follow, and the language is clear even for unseasoned researcher to understand. Though, felt the introduction lacked details about the purpose and background of the review. I would recommend that the authors add more information and literature reviews to expand how the review would improve the use of common reference group in analysis of epidemiologic studies.

Overall, the paper is sound and would make an interesting addition to literature to augment epidemiological research.

I do not have the expertise to comment on the statistical approaches on this paper.

6. PLOS authors have the option to publish the peer review history of their article (what does this mean?). If published, this will include your full peer review and any attached files.

Reviewer #1: No

---

## [Author Response · Author response to Decision Letter 0]

14 Sep 2023

Dear Dr. Akolo,

We thank you and the reviewer for your kind reviews of our manuscript, which have helped improve the clarity of our manuscript.

We have addressed all comments you and the reviewer have made, which are detailed in our Response to the Editor and the Reviewer.

I look forward to hearing from you at your earliest convenience.

Respectfully,

Bao-Ping Zhu, MD, PhD, MS

Director, Office of Science Quality and Library Services

Office of Science

United States Centers for Disease Control and Prevention

---

## [Decision Letter · Decision Letter 1]

4 Dec 2023

Untangling the Effects of Multiple Exposures with a Common Reference Group in an Epidemiologic Study: A Practical Revisit

PONE-D-23-13611R1

Dear Dr. Bao-Ping Zhu,

We’re pleased to inform you that your manuscript has been judged scientifically suitable for publication and will be formally accepted for publication once it meets all outstanding technical requirements.

Kind regards,

Awatif Abid Al-Judaibi, PhD

Academic Editor

PLOS ONE

Reviewer's Responses to Questions

**Comments to the Author**

1. If the authors have adequately addressed your comments raised in a previous round of review and you feel that this manuscript is now acceptable for publication, you may indicate that here to bypass the “Comments to the Author” section, enter your conflict of interest statement in the “Confidential to Editor” section, and submit your "Accept" recommendation.

Reviewer #1: All comments have been addressed

Reviewer #2: All comments have been addressed

2. Is the manuscript technically sound, and do the data support the conclusions?

Reviewer #1: Yes

Reviewer #2: Yes

3. Has the statistical analysis been performed appropriately and rigorously? 

Reviewer #1: I Don't Know

Reviewer #2: Yes

4. Have the authors made all data underlying the findings in their manuscript fully available?

Reviewer #1: Yes

Reviewer #2: Yes

5. Is the manuscript presented in an intelligible fashion and written in standard English?

Reviewer #1: Yes

Reviewer #2: Yes

6. Review Comments to the Author

Reviewer #1: (No Response)

Reviewer #2: The authors have integrated the reviewers feedback well. The introduction is clear and provides arguments as to why the CRG approach might be appropriate.

The examples used are understandable and the data is provided in the supplementary material.

The authors also discussed possible disadvantages to performing the CRG which is very important for studies with a small sample size.

There is one type in the Discussion section line 236 "requring" change to "requiring".

7. PLOS authors have the option to publish the peer review history of their article (what does this mean?). If published, this will include your full peer review and any attached files.

Reviewer #1: No

Reviewer #2: No

---

## [Editor Report · Acceptance letter]

8 Dec 2023

PONE-D-23-13611R1 

Untangling the Effects of Multiple Exposures with a Common Reference Group in an Epidemiologic Study: A Practical Revisit 

Dear Dr. Zhu:

I'm pleased to inform you that your manuscript has been deemed suitable for publication in PLOS ONE. Congratulations! Your manuscript is now with our production department. 

Kind regards, 

on behalf of

Professor Awatif Abid Al-Judaibi 

Academic Editor

PLOS ONE